# ISTD-PDS7: A Benchmark Dataset for Multi-Type Pavement Distress Segmentation from CCD Images in Complex Scenarios

**Weidong Song [1], Zaiyan Zhang [1,2,\*], Bing Zhang [1], Guohui Jia [3], Hongbo Zhu [1] and Jinhe Zhang [1]**

1   School of Geomatics, Liaoning Technical University, Fuxin 123000, China
2   College of Mining Engineering, Heilongjiang University of Science and Technology, Harbin 150022, China
3   School of Resources and Civil Engineering, Liaoning Institute of Science and Technology, Benxi 117004, China
*   Correspondence: 2014800205@usth.edu.cn; Tel.: +86-1894-600-7768

**Abstract:** The lack of large-scale, multi-scene, and multi-type pavement distress training data reduces the generalization ability of deep learning models in complex scenes, and limits the development of pavement distress extraction algorithms. Thus, we built the first large-scale dichotomous image segmentation (DIS) dataset for multi-type pavement distress segmentation, called ISTD-PDS7, aimed to segment highly accurate pavement distress types from natural charge-coupled device (CCD) images. The new dataset covers seven types of pavement distress in nine types of scenarios, along with negative samples with texture similarity noise. The final dataset contains 18,527 images, which is many more than the previously released benchmarks. All the images are annotated with fine-grained labels. In addition, we conducted a large benchmark test, evaluating seven state-of-the-art segmentation models, providing a detailed discussion of the factors that influence segmentation performance, and making cross-dataset evaluations for the best-performing model. Finally, we investigated the effectiveness of negative samples in reducing false positive prediction in complex scenes and developed two potential data augmentation methods for improving the segmentation accuracy. We hope that these efforts will create promising developments for both academics and the industry.

**Keywords:** pavement CCD images; deep learning; distress semantic segmentation; ISTD-PDS7

## 1. Introduction

The automatic detection of road pavement distress is necessary to realize the maintenance and monitoring of complex traffic networks, and is an effective way to improve the quality of road service [1]. For both cement concrete and asphalt pavements, in the actual operation process, unidirectional cracks, alligator cracks, broken slabs, potholes, and other types of distress can appear under the comprehensive influence of the traffic volume, load, temperature, moisture, and weathering, and are collectively called pavement distress [2]. Pavement distress accelerates highway aging, greatly reduces driving comfort, increases vehicle wear, and can increase avoidance actions that may lead to collisions, posing potential threats to highway and driving safety [3]. According to a survey conducted by the Ministry of Transport of China, the road maintenance mileage in China has reached 5.28 million kilometers, which is about 99.4% of the total road mileage in 2021 [4]. Thus, with the increasing demand for road maintenance, computer vision-based road condition assessments have become a research hotspot in the industry.

The traditional manual inspection approach is time-consuming, laborious, and bi-ased. To solve this problem, optical imaging with onboard charge-coupled device (CCD) sensors combined with digital image processing technologies has attracted much attention because it can automatically monitor pavement conditions [5]. For several decades, researchers have been working on the application of computer vision technologies for pavement distress assessment. Early studies based on digital image processing technologies [6–10]

and machine learning technologies [11–14] are extensive [15], but their flaws are obvious as these are affected by factors, such as the environment, traffic load, and maintenance conditions. There are three obvious characteristics in the CCD distress images of highway pavements in natural scenes: (1) the image quality is greatly affected by the light intensity, highway dryness, shadow, and other interference; (2) the background is complex and changeable, there is strong speckle noise, and a low target signal-to-noise ratio (SNR); and (3) there are many distress classes, complex topological structures, and the gray feature difference is very small. For example, as shown in Figure 1, the impulse noise brought by the grain-like pavement texture breaks the crack and undermines its continuity, and shadows reduce the contrast between the crack and the background. As a result, these traditional methods struggle to achieve complete multi-type pavement distress segmentation from complex backgrounds [16,17].

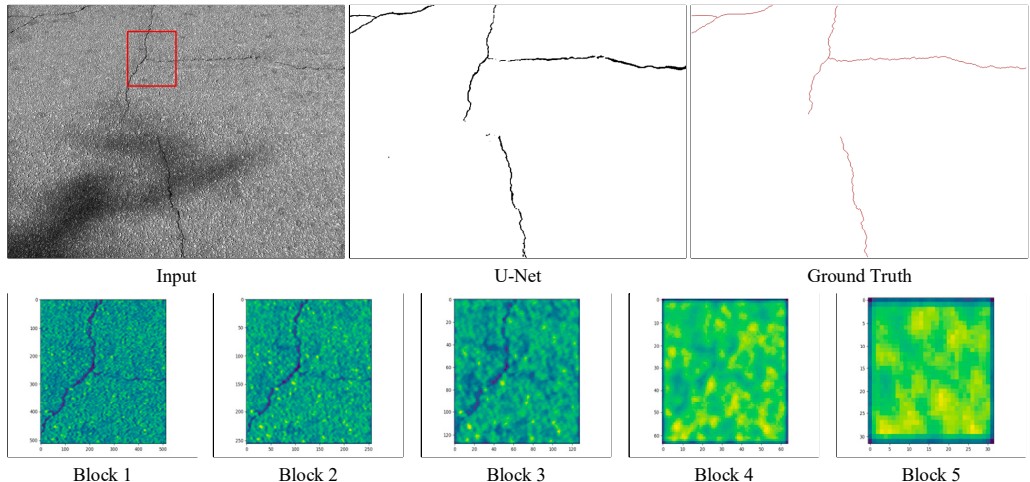

**Figure 1.** A real example of distress segmentation using U-Net. The second row shows the feature maps of blocks with different depths in U-Net (for the image patch denoted by the rectangle in the input image).

Recent studies have shown that, with the introduction of deep learning (DL) models in the fields of photogrammetry, remote sensing, and computer vision, DL based methods are now performing a dominant role in the detection of road surface distress with CCD images [18–20]. Studies, such as [16,21–23], have shown that deep convolutional neural networks (DCNNs) can be used for the automatic segmentation of pavement cracks. These deep architectures build high-level features from low-level primitives by hierarchical convolution of the input. As can be seen in the second row of Figure 1. These DCNN architectures can build high-level features from low-level primitives by hierarchically convolving the sensory inputs. Typically, DL based crack detection methods can be classified into three categories: (1) image classification based methods, (2) object detection based methods, and (3) semantic segmentation based methods [20].

However, it must be noted that, compared with classification [24–26] and detection [27–29], semantic segmentation can provide more accurate geometric target description for a wide range of applications, such as geometric feature measurement of distress [2,30], severity division [21], and quantitative assessment of pavement conditions [16], which is also the research focus of this paper. In addition, most studies have focused on only one or two types of distress. Ouma's research focused on linear crack detection [31] and Siriborvornratanakul's re-search focus was on crater detection [32].

The tasks and datasets for image segmentation are closely related in the deep learning era. Some of the segmentation tasks, such as those considered in [33–36], are even directly built upon the datasets. However, most of the pavement data collection systems are complex and complicated, and it is costly to label this amount of data [23]. As shown in Table 1, the datasets published so far do often consist of less than 500 pavement distress images [37–40].

Moreover, most of the available pavement distress datasets have been collected from highways, which are regularly maintained. As a result, the collected pavement datasets contain images with less variety of pavement distress [23], fewer scenes, and almost all the datasets are derived from images of asphalt road. Therefore, a bench-mark dataset is urgently required to address these challenging problems. Previous studies have found that the multiple distress types have complex topological structures, different sizes, and similar gray-level values, and labeling experts commonly disagree on the accurate classification of the categories [22]. Therefore, the construction of a multi-type pavement distress dataset according to the conventional semantic segmentation task is not conducive to accurate segmentation. Recent research has found that a category agnostic dichotomous image segmentation (DIS) task defined on non-conflicting annotations can be used to accurately segment objects with different structural complexities regardless of their characteristics [40].

**Table 1.** Comparison of the established publicly available datasets for pavement distress segmentation. AP and CCP are the abbreviations for asphalt pavement and cement concrete pavement, respectively.

| Task | Dataset | Illumination | Proportion of Pavement Type/% | Equipment |
|---|---|---|---|---|
| Crack segmentation | CrackLS315 [30] | Laser | AP: 100.0% | Area-array camera |
| | CRKWH100 [30] | Visible light | AP: 100.0% | Linear-array camera |
| | CrackTree260 [30] | Visible light | AP: 100.0% | Area-array camera |
| | AigleRN [39] | Visible light | AP: 100.0% | Area-array camera |
| | CFD [37] | Visible light | CCP: 1.7%; AP: 98.3% | Smartphone |
| | CRACK500 [38] | Visible light | CCP: 2.4%; AP: 97.6% | Smartphone |
| | GAPs384 [38] | Laser | AP: 100.0% | Linear-array camera |
| Multi-type distress DIS | ISTD-PDS7 | HID lamp | CCP: 29.4%; AP: 70.6% | Area-array camera |

Inspired by these observations, in this study, we built a highly detailed DIS dataset, named ISTD-PDS7, for multi-type pavement distress segmentation. The dataset has a sufficient scale, labeling precision, and scene diversity. We hope that the ISTD-PDS7 dataset can contribute to improving the robustness and reliability of automatic pavement distress extraction algorithms in complex scenarios, and further promote the research progress into automatic assessment of large-scale highway pavement conditions. The ISTD-PDS7 dataset and benchmarks will be made publicly available at: https://ciigis.lntu.edu.cn/. The main contributions of this paper can be summarized as the following three aspects:

1. A large-scale extendable DIS dataset—ISTD-PDS7—containing 18,527 CCD images and 7 types of pavement distress, was built and annotated manually by 4 experts in the field of pavement distress detection. Finally, highly detailed binary segmentation masks were generated. The dataset was analyzed in detail from three aspects: image dimension, image complexity, and annotation complexity.

2. Based on the new ISTD-PDS7 dataset, we compared the cutting-edge segmentation models with different network structures and made a comprehensive evaluation and analysis of their pavement distress segmentation performance. These results will serve as the baseline results for future works.

3. We briefly review the numerous previously published datasets. We also describe the detailed evaluation and comparative experiments conducted between these datasets and ISTD-PDS7, and propose their comparison results in crack segmentation.

The rest of this paper is organized as follows: A brief review of the related work is provided in Section 2. Section 3 presents the details of the dataset collection and labeling, data analysis, and splitting of the dataset. Subsequently, we provide a description of the baseline algorithms for the benchmark evaluation in Section 4. Section 5 first describes the implementation details and the dataset setup of this study, then a performance analysis of the baseline algorithms on ISTD-PDS7 is given. We also evaluate and compare the crack segmentation performance obtained with the different public datasets and the new ISTD-PDS7 dataset, in addition to the impact of negative samples. Finally, our conclusions are given in Section 6.

## 2. Related Work

### 2.1. Automated Crack Detection

Crack detection has always been a research hotspot in the field of automatic distress detection. In 2016, Zhang et al. [41] used smartphones to collect highway images to build a dataset, and LeNet-5 [42] was used for crack detection for the first time. Subsequently, a pre-trained VGG-16 model based on ImageNet was transferred to the crack detection task for asphalt and cement pavements by Gopalakrishnan et al. [43] and achieved a superior crack detection performance. However, the above two methods need to be converted into a fully convolutional network to obtain the image segmentation results. Therefore, crack segmentation based on an encoder–decoder network structure is becoming more common. In view of the excellent performance of the U-Net [44] architecture in the field of biomedical image segmentation, Jenkins et al. [45] developed a semantic segmentation algorithm for pavement cracks based on U-Net. However, due to the lack of data (80 training images and 20 validation images), the generalization ability of the model was insufficient. Researchers have since made a variety of improvements to the network structure based on the U-Net architecture. For example, Lau et al. [46] replaced the encoder phase of U-Net with a pre-trained ResNet-34 model, and the method achieved an F1-score of 96% and 73% on the CrackForest dataset (CFD) [37] and Crack500 dataset [38], respectively. Escalona et al. [47] implemented three different U-Net models for crack segmentation, and performed segmentation performance tests on the CFD [37] and AigleRN [39] datasets. Furthermore, in order to balance the segmentation efficiency and accuracy, Polovnikov et al. [48] proposed a lightweight U-Net-based network architecture called DAUNet, which was tested on the publicly available datasets, and was found to be able to effectively detect cracks in complex scenes.

In addition, Zou et al. [30] improved the SegNet [49] architecture and proposed an end-to-end trainable DeepCrack network to detect cracks where the pavement image pixels are distinguished into crack and non-crack background forms. In addition, Fan et al. [50] proposed a novel highway crack detection algorithm based on DL and adaptive image segmentation. Xu et al. [51] proposed a new network architecture, an enhanced high-resolution semantic network (EHRS-Net), which was suitable for tiny cracks and noised pavement cracks. In order to compensate the transformer for the deficiency of local features, Xu et al. [20] proposed a new network for pavement crack detection from CCD images, called LETNet, which has strong robustness.

### 2.2. Multi-Type Distress Segmentation Approaches

However, cracks are not the only type of distress in pavements and some researchers have recently turned their attention to multi-type distress segmentation. For example, Lõuk et al. [52] applied a U-Net-like network architecture with different context resolution levels to integrate more contextual information. The pavement distress detection system (PDDS) proposed by Lõuk et al. [52] can output multi-class pavement distress regions, and they have stated that their future research work will focus on the 11 defect categories defined by the Estonian Road Administration. Majidifard et al. [16] developed a method based on the combination of U-Net and YOLO, which can effectively distinguish cracks, dense cracks, and potholes. In addition, Zhang et al. [23] collected urban highway pavement images in Montreal, Canada, and produced semantic segmentation datasets for potholes, patches, lane lines, unidirectional cracks, and network cracks. The authors also proposed and evaluated a method for the automatic detection and classification of pavement distress classes using a convolutional neural network (CNN) and low-cost video data, where the detection rate and classification accuracy of the model both reached 83.8%.

It must be noted that DL is a data-driven technology, and massive labeled data can effectively reduce the risk of overfitting and improve the generalization performance of a model. However, none of the above studies proposed a model based on a comprehensive dataset which covers all the highway pavement distress classes under different conditions in natural scenes [16]. In addition, due to the lack of standardized images as a test set, it is

difficult to measure the geometric features and evaluate the accuracy of pavement distress segmentation results.

### 2.3. Existing Datasets for Pavement Distress Segmentation

Although many different distress detection methods have been proposed to date, there is still a lack of public datasets that are both large enough and annotated in a standardized manner. The datasets released to date typically contain fewer than 500 images, e.g., CrackTree260, CrackLS315, and CrackWH100 were annotated manually at a single-pixel width [30]. CFD [37], CRACK500 [38], and AigleRN [39] were annotated manually according to the actual crack width. See Table 1 for details.

- **CrackTree260** [30]: The CrackTree260 dataset consists of 260 pavement crack images with a size of 800 × 600 pixels, for which the pavement images were captured by an area-array camera under visible light illumination.
- **CrackLS315** [30]: The CrackLS315 dataset contains 315 road pavement images captured under laser illumination. These images were captured by a linear-array camera, at the same ground sampling distance.
- **CRKWH100** [30]: The CRKWH100 dataset contains 100 road pavement images captured by a linear-array camera under visible light illumination. The linear-array camera captures the pavement at a ground sampling distance of 1 mm.
- **CFD** [37]: The CFD dataset consists of 118 iPhone 5 images of cracks in the urban pavement of Beijing in China, where each image is manually labeled with the ground-truth contour and the size is adjusted to 480 × 320 pixels. The dataset also includes a few images that are contaminated by small oil spots and water stain noise.
- **Crack500** [38]: The Crack500 dataset consists of 500 2000 × 1500 pixel crack images, with a few containing oil spots and shadow noise.
- **AigleRN** [39]: The AigleRN dataset contains 38 preprocessed grayscale images of pavements in France, with the size of one half of the AigleRN dataset being 991 × 462 pixels, and the size of the other half being 311 × 462 pixels.
- **GAPs** [53]: In 2017, a freely available large pavement distress detection dataset called the German Asphalt Pavement (GAPs) dataset was released by Eisenbach et al. [51], which has since received considerable attention from several research groups (e.g., [54–56]). The GAPs dataset was the first attempt at creating a standard benchmark pavement distress image dataset for DL applications. It includes 1969 grayscale pavement images (1418 for training, 51 for validation, and 500 for testing) with various distress types, including cracks (longitudinal/transverse, alligator, sealed/filled), potholes, patches, open joints, and bleeding [53]. Unfortunately, the method of bounding box annotation is not very friendly for semantic segmentation tasks. To solve this problem, Yang et al. [38] selected and annotated 384 crack images from the GAPs dataset at the pixel level and built a new segmented dataset called GAPs384. It is worth noting that all the images in the GAPs dataset were collected from the pavements of three different German federal highways. The shooting conditions were dry and warm, so the GAPs dataset is suitable for studying the segmentation and extraction problems of pavement distress in urban highways and expressways with good highway conditions. More recently, Stricker et al. [22] released the publicly available GAPs-10 m dataset for semantic segmentation. This dataset contains 20 high-resolution images (5030 × 11505 pixels, each corresponding to 10 m of highway pavement) that cover 200 m of asphalt roads with different asphalt surface types and a wide variety of distress classes [22]. The corresponding multi-class distress labels were annotated by experts; this dataset is currently the only publicly available dataset with high-resolution images.
- **Others:** Although some of the larger datasets recently published, such as the dataset made up of 700K Google Street View images [57] or the Global Road Damage Detection Challenge (GRDDC) 2020 dataset [58], are mostly used for object detection tasks in

severely damaged images as they do not have the resolution level required for highway damage condition assessment.

In summary, although automatic pavement distress detection algorithms based on DL have made good progress in recent years, the existing publicly available datasets for pavement distress segmentation still have the problems of a small scale, few scenes, single pavement type, low resolution, and unmeasurable segmentation results. Therefore, a benchmark dataset is urgently required to address these challenging problems.

## 3. ISTD-PDS7 Dataset

### 3.1. Data Collection and Annotation

**Data Collection:** To solve the data problem (see Section 2) based on complex scenes and high-resolution CCD images, we built a detailed DIS dataset for multi-type pavement distress segmentation named ISTD-PDS7. The original images of the ISTD-PDS7 dataset were acquired using a mobile acquisition vehicle (see Figure 2).

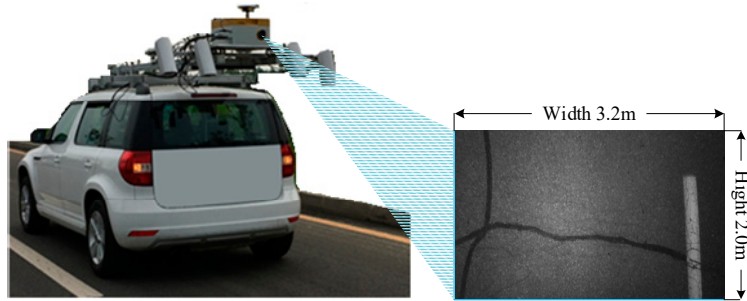

**Figure 2.** Mobile pavement distress detection equipment. The vehicle was equipped with an acA4096 array camera and two hernia lamps to reduce the influence of light intensity. The image resolution was 3517 × 2193 pixels, and the single pixel size was 0.91 × 0.91 mm.

In this study, we first collected the original images from 504 roads in different regions of China and manually screened 30,000 original CCD images based on four pre-designed keywords according to the extent of pavement damage: high-quality asphalt pavement, low-quality asphalt pavement, high-quality cement concrete pavement, and low-quality cement concrete pavement. Then, we developed a lossless cropping tool, and it took four experts six months to crop the seven kinds of distress (transverse crack, longitudinal crack, cement concrete crack, alligator network crack, broken slabs, patch, and pothole) and negative sample areas in the earlier screened images. Finally, 18,527 sample images were obtained according to the nine kinds of complex scene (clear pavement, fuzzy pavement, bright light, weak light, dry pavement, slippery pavement, shadow, stain, and sundries) for each distress type, covering 6553 distress sample images and 11,974 negative sample images with interference noise (Figure 3), which can effectively reduce the false extraction of complex interference noise (we illustrate this point in the experiments). Note that the selection and tailoring strategy was similar to the approach of Everingham et al. [59] and Zou et al. [30]. The distressed area typically occupies a small proportion of the whole image, and many background areas have no practical significance for the training process. Therefore, most of the selected distress-containing images contain only a single target to provide rich and highly detailed structures. Meanwhile, the segmentation and labeling confusion caused by the co-occurrence of multiple distress types from different categories is avoided as much as possible. Specifically, the selection criteria for the distress images can be summarized as follows:

- We covered more categories while reducing the number of "redundant" samples with simple structures that are already included in the other existing datasets. The focus was on increasing the scene richness of each type of distress sample, which is crucial for improving the reasoning ability of the network model. As shown in Figure 3, we

screened seven types of distress sample images, where each type of distress covered nine types of complex scenes (Figure 4).

- We enlarged the intra-type dissimilarities of the selected distress types by adding more diversified intra-type images (see Figure 4). First, the same distress type may appear with different lengths, widths, and topologies, due to the diversity of the distress formation mechanisms in rural pavements. Second, the appearance of road pavement distress is greatly affected by dust accumulation and humidity. For example, the black appearance of the crack and the white appearance of the crack shown in the first row of Figure 4. Finally, the vibration during the shooting also affects the clarity of the distress in the imagery.

- We included more images that are highly similar to the road surface distress in terms of gray-level and texture characteristics, which are called negative samples, such as shadows, water or oil stains, dropped objects, pavement appendages, etc. (Figure 5). These are common in actual distress detection tasks, but they are ignored by the other datasets due to their complex types or collection difficulties.

The purpose of the DIS task is to obtain accurate pixel regions of the different distress types to analyze the condition of the pavement, which seems to be contradictory with the image collection based on pre-designed keywords/types in this paper. The main reasons for this include: (1) To facilitate image retrieval and organization in the construction of the large-scale pavement distress dataset. (2) Collecting samples according to the distress types is a reasonable way to ensure the characteristics (such as texture, topological structure, contrast, background complexity, etc.) of the distress sample diversity, which can improve the robustness and generalization of type-agnostic segmentation. (3) Prior to the development of the different pavement distress inspection systems, the existing datasets needed to be reorganized and extended according to the task requirements. The type information provided in this paper will help developers to quickly track down the required samples. Therefore, the type-based collection approach is intrinsically consistent with the objective of the pavement distress DIS task.

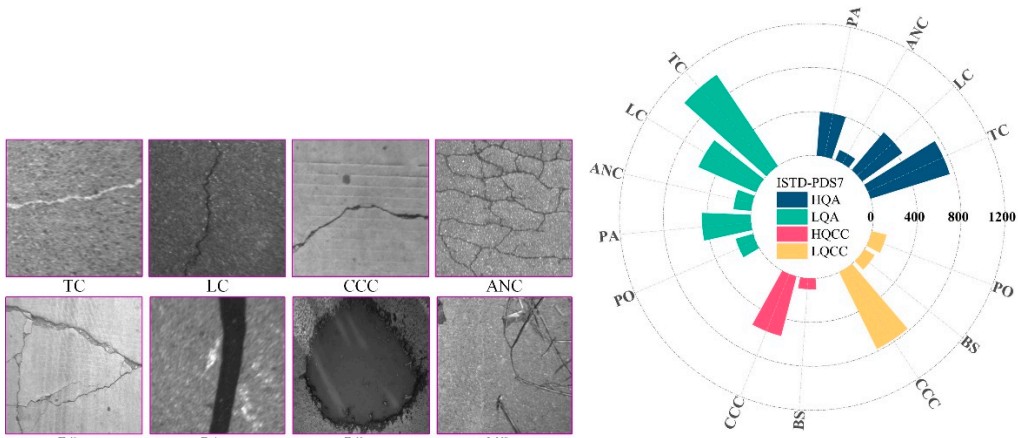

**Figure 3.** (**Left**) example images of ISTD-PDS7, where "TC", "LC", "CCC", "ANC", "BS", "PA", "PO", and "NS" are the abbreviations for "transverse crack", "longitudinal crack", "cement concrete crack", "alligator network crack", "broken slab", "patch", "pothole", and "negative sample". (**Right**) distress types and groups of the ISTD-PDS7 dataset, where "HQA", "LQA", "HQCC", and "LQCC" are the abbreviations for high-quality asphalt pavement, low-quality asphalt pavement, high-quality cement concrete pavement, and low-quality cement concrete pavement.

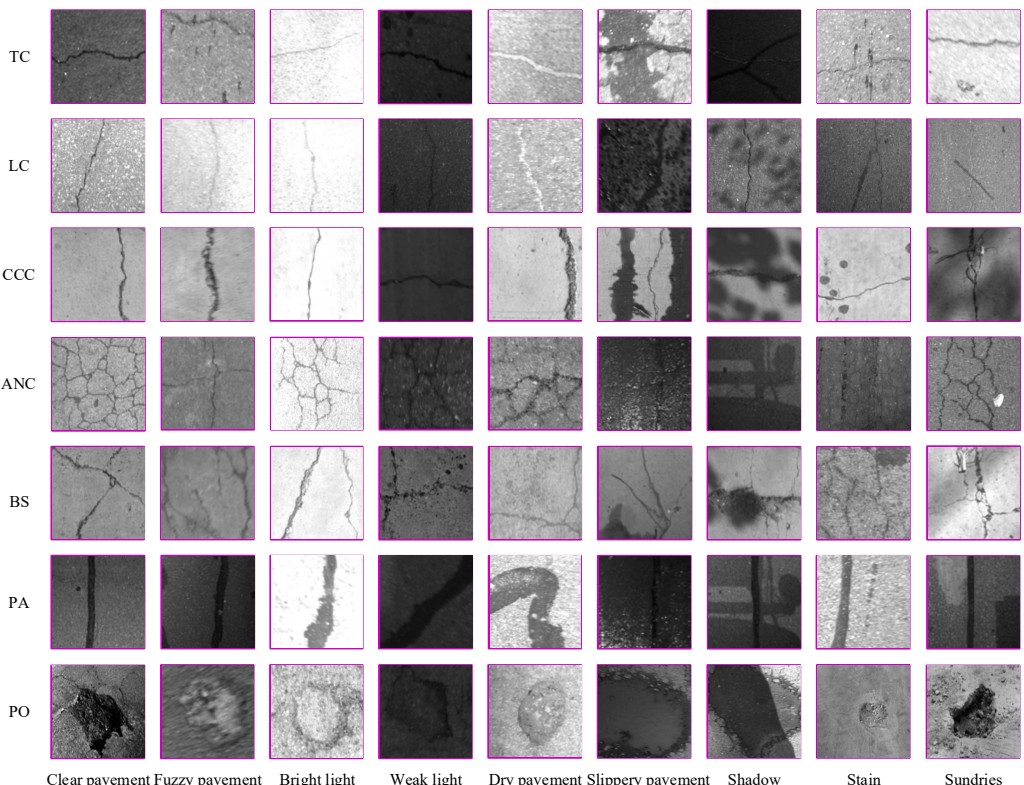

**Figure 4.** Samples from the ISTD-PDS7 dataset: nine scenes of each distress class are shown. There are 6533 images within seven classes.

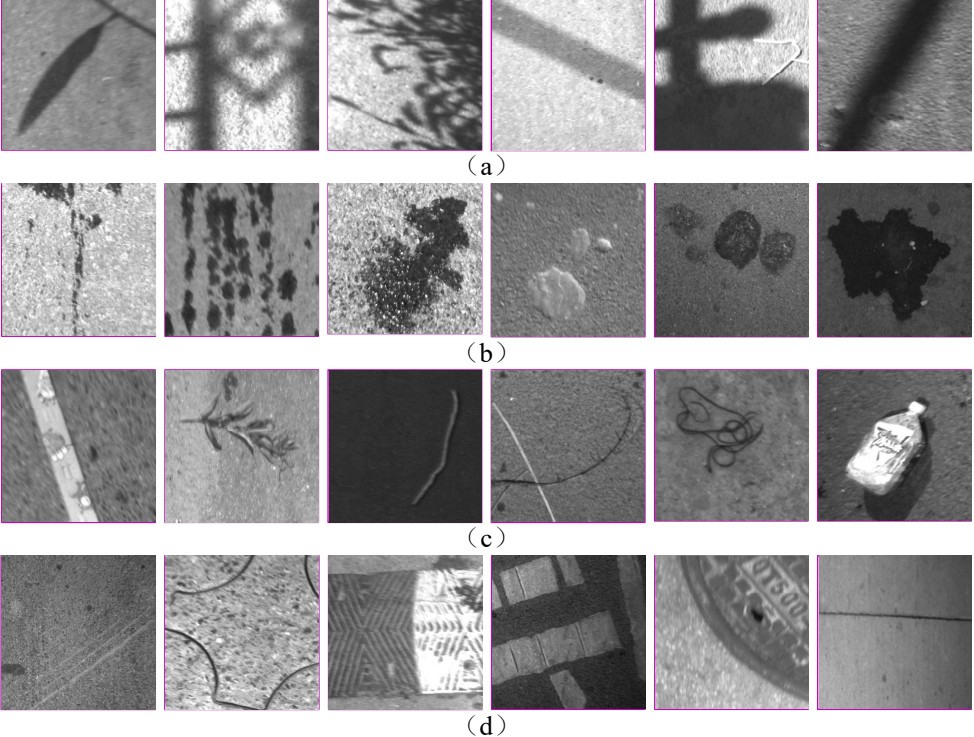

**Figure 5.** Negative samples from the ISTD-PDS7 dataset. (**a**) Under the influence of interference, such as trees, light poles, and acquisition equipment, shadows appear in pavement images, leading to

deviations in the overall gray-level distribution of the image. Some shadows have similar structural characteristics to the patch, but the roughness of the patch is greater than that of strip shadows; (**b**) Water stains, oil stains, mud stains, and other interference noise, some of which are similar to the linear characteristics of cracks, and some are similar to the appearance of potholes; (**c**) Litter, such as branches, weeds, and garbage, has obvious edge characteristics; (**d**) Other pavement interference noise: scratches, cutting marks, speed bumps, zebra crossings, manhole covers, and cement pavement expansion joints. The existence of negative samples further increases the challenge of the dataset and helps to evaluate the robustness and generalization of different distress detection models.

**Dataset Annotation:** Each image was manually annotated with pixel-wise precision by four pavement distress detection field experts using LabelMe (Figure 6). The average labeling time for each image was about 20 min and some alligator network crack images took up to 1 h. Figure 6a–g shows the different levels of details in annotation and the differences in detail labeling during the ISTD-PDS7 dataset and the existing public datasets in terms of crack labeling. Figure 6h shows annotated samples of seven types of pavement distress and negative samples in complex scenes. Figure 6i demonstrates the greater diversity of the intra-type structure complexities of the ISTD-PDS7 dataset.

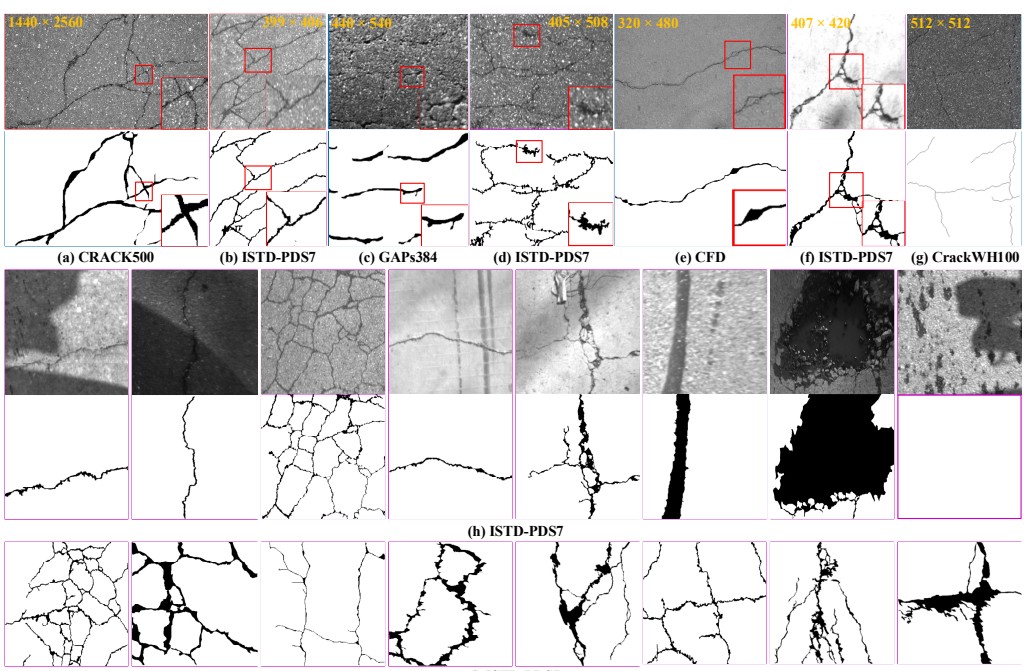

**Figure 6.** Qualitative comparison of the different datasets: (**a**–**d**) indicate that ISTD-PDS7 provides more detailed crack labeling; (**e**,**f**) show the differences in detail labeling between the CFD [37] dataset and the ISTD-PDS7 dataset; (**g**) is a sample of the CRACKWH100 dataset [30], which was annotated manually with a single-pixel width; (**h**) shows labeling cases of seven types of distress and negative samples in complex scenarios; and (**i**) demonstrates the structural complexity and diversity of the alligator network cracks within the ISTD-PDS7 dataset.

### 3.2. Data Analysis

- For a deeper insight into the ISTD-PDS7 dataset, we compared the dataset with seven other related datasets: three crack segmentation datasets annotated by a single-pixel width, i.e., CrackLS315, CrackWH100, and CrackTree260 [30], and four datasets annotated manually by the actual width of the crack, i.e., AigleRN [39], CFD [37], CRACK500 [38], and GAPs384 [38]. The comparison was made mainly from the four metrics of image number, image dimension, image complexity, and annotation complexity, and they are described as follows: **Image dimension** is crucial to the

segmentation task, as it directly affects the accuracy, efficiency, and computational cost of the segmentation [40].

- **Image Complexity** is described by the image information entropy ($IE$), which can quantitatively represent the difficulty of object recognition or extraction in complex scenes [60]. The $IE$ ($IE = \sum_{i=0}^{i=255} P_i \log_2 P_i$), where $P_i$ represents the proportion of pixels whose gray value is $i$ in the image measures the information contained in the aggregation features of the gray-level distribution in an image from the perspective of information theory. The greater the $IE$ in an image, the more information the image contains [61,62].

- **Annotation Complexity** is described by the three metrics of the isoperimetric inequality quotient ($IPQ$) [63–65], the number of distress contours ($Cnum$), and the number of points marked ($Pnum$). Among these metrics, $IPQ = L^2/4\pi A^2$, where $L$ and $A$ denote the distress perimeter and the region area, respectively. The $IPQ$ represents the structural complexity of the labeled distress types. The $Cnum$ is the number of closed contours involved in the labeling, which can quantitatively reflect the complexity of the topological structure of the distress. The $Pnum$ metric is the number of labeling points needed to delineate the outline of the distress example [66], which can quantitatively reflect the fineness of the labeling and the labor cost.

Table 2 lists the statistical findings for four indicators across various data sets. Note that red denotes the best outcomes, green the second-best outcomes, blue the third-best outcomes, and negative samples are excluded from labeling complexity statistics. In addition, these four metrics are complementary and can provide a comprehensive analysis of the complexity of the original imagery and annotated objects, see Figure 7 for details.

The mean values ($H$, $W$, $D$) and standard deviations ($\sigma H$, $\sigma W$, $\sigma D$) of the image height, width, and diagonal length of each dataset are listed in Table 2. The CRACK500 dataset has the largest average image dimensions, but it only contains 500 images. In view of the distress area typically occupying a small proportion of the whole CCD image, and many background areas having no practical significance for the training process, targeted cropping was carried out on the distress area in the ISTD-PDS7 dataset, so that the average image dimension of ISTD-PDS7 is relatively small. In addition, the targets of the seven open-source datasets are primarily cracks, which limits their application in diversified tasks. From the mean value and standard deviation of the $IE$ in Table 2 and Figure 7a, it is apparent that, compared with the other public datasets, the ISTD-PDS7 dataset has the highest image complexity. Figure 6h also intuitively indicates that ISTD-PDS7 is closer to the actual application scenario.

**Table 2.** Data analysis of the existing datasets.

| Task | Dataset | Number | Image Dimension | | | Image Complexity | Annotation Complexity | | | Annotation Method |
|------|---------|--------|------|------|------|------|------|------|------|------|
| | | *I* num | $H$ $\pm\sigma H$ | $W$ $\pm\sigma W$ | $D$ $\pm\sigma D$ | $IE$ $\pm\sigma IE$ | $IPQ$ $\pm\sigma IPQ$ | $Cnum$ $\pm\sigma Cnum$ | $Pnum$ $\pm\sigma Pnum$ | |
| Crack | CrackLS315 [6] | 315 | 512.00 ± 0.00 | 512.00 ± 0.00 | 724.00 ± 0.00 | 50.12 ± 11.46 | 297.70 ± 199.48 | 3.40 ± 2.60 | 618.29 ± 415.57 | Single-pixel width |
| | CrackWH100 [6] | 100 | 512.00 ± 0.00 | 512.00 ± 0.00 | 724.00 ± 0.00 | 36.72 ± 9.15 | 432.68 ± 452.24 | 2.90 ± 4.27 | 855.45 ± 887.29 | |
| | CrackTree260 [6] | 260 | 624.92 ± 48.77 | 833.23 ± 65.03 | 1041.54 ± 81.29 | 53.45 ± 12.15 | 1122.18 ± 996.63 | 8.64 ± 16.07 | 2551.08 ± 2195.26 | |
| | AigleRN [38] | 38 | 522.35 ± 151.02 | 692.05 ± 280.26 | 890.23 ± 244.33 | 32.18 ± 9.27 | 437.02 ± 416.94 | 17.40 ± 14.62 | 1374.35 ± 1014.11 | Actual width |
| | CFD [36] | 118 | 320.00 ± 0.00 | 480.00 ± 0.00 | 577.00 ± 0.00 | 36.42 ± 9.36 | 106.60 ± 56.30 | 3.69 ± 4.19 | 661.78 ± 457.74 | |
| | CRACK500 [37] | 500 | 1568.38 ± 313.60 | 2594.61 ± 240.45 | 3042.13 ± 303.57 | 59.66 ± 13.21 | 91.24 ± 95.42 | 18.11 ± 25.92 | 3603.03 ± 2065.43 | |
| | GAPs384 [37] | 384 | 551.65 ± 99.43 | 540.00 ± 0.00 | 775.15 ± 69.60 | 55.29 ± 12.19 | 48.44 ± 40.89 | 5.26 ± 5.57 | 452.22 ± 286.27 | |
| DIS | ISTD-PDS7 | 18527 | 375.47 ± 99.50 | 371.00 ± 114.44 | 529.45 ± 145.73 | 89.28 ± 13.63 | 134.82 ± 265.07 | 9.10 ± 16.95 | 1083.38 ± 1054.07 | |

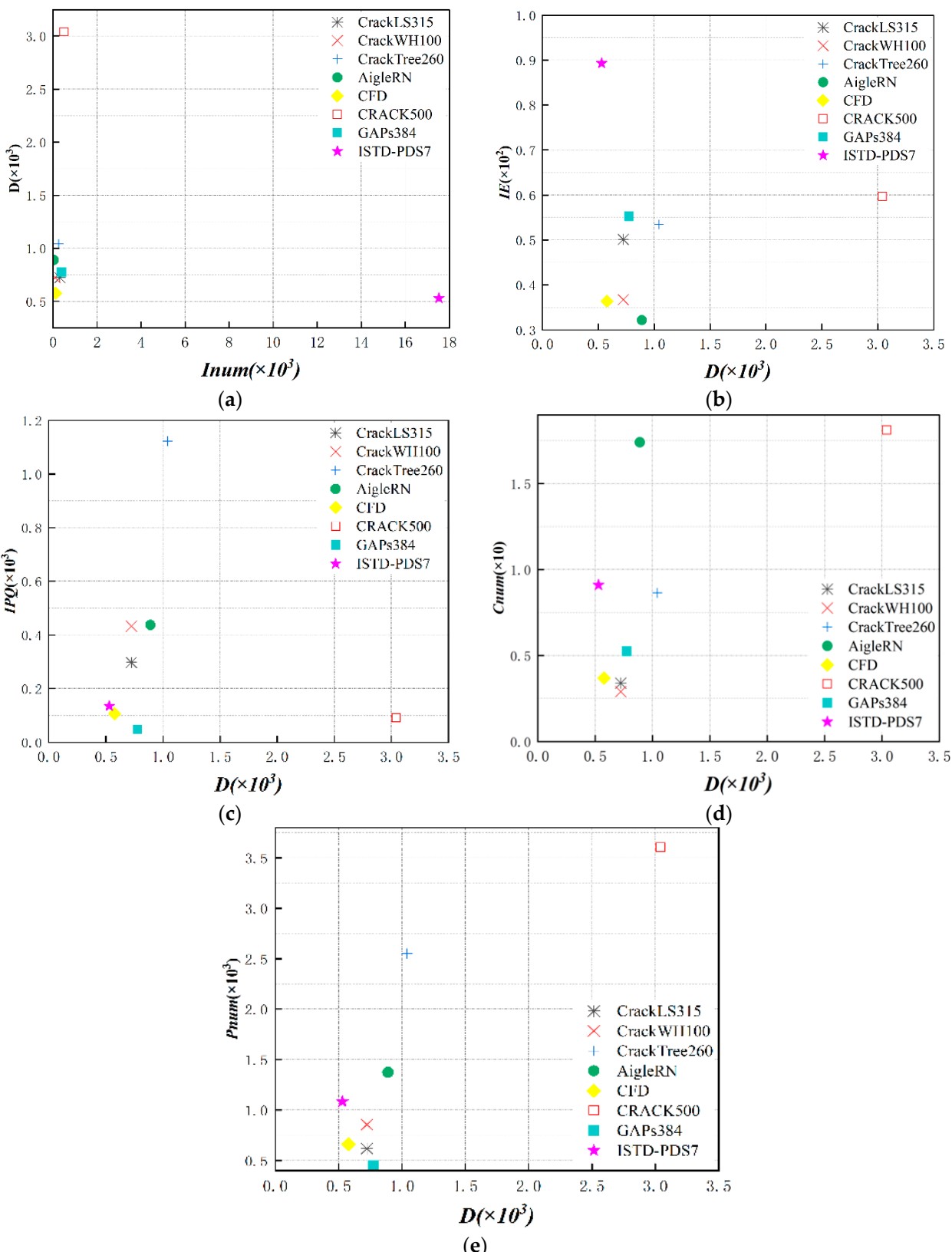

**Figure 7.** Correlations between the different complexity metrics. These four complexity measurements are complementary and can provide a comprehensive analysis of the complexity of the original imagery and annotated objects (**a**–**e**).

As shown in Table 2 and Figure 7, the CrackTree260 dataset achieves the highest complexity in terms of the average structural complexity *IPQ*, but its *Inum* and *IE* values are far lower than for the ISTD-PDS7 dataset. In addition, although the ISTD-PDS7 dataset contains simply shaped patches (single contour), potholes (single contour), and negative sample labels (no contour), the mean and standard deviation of *Cnum* still achieve high scores, suggesting that the cracks in the new dataset are a more refined structure composed of multiple contours. The average *Pnum* of the ISTD-PDS7 dataset is over 1000, which is around two times more complicated than the GAPs384 dataset. This shows that the ISTD-PDS7 dataset provides more detailed annotation on images with smaller dimensions than the other datasets.

### 3.3. Dataset Splitting

As shown in Table 3, we split the 18,527 images in the ISTD-PDS7 dataset into three subsets: ISTD-TR (15,774), ISTD-VD (1753), and ISTD-TE (1000) for training, validation, and testing, respectively. The distress types in ISTD-TR and those in ISTD-VD and ISTD-TE are mainly consistent. In addition, using the complexity of the scene and topology structure as the screening index, the 1000 images of ISTD-TE are further split into a subset, containing 532 crack images and 18 negative samples with texture similarity noise, which is called ISTD-CRTE, to evaluate the segmentation performance for pavement cracks. Overall, the ISTD-PDS7 dataset is designed to meet the challenges of model training and performance evaluation in complex scenarios.

**Table 3.** Dataset splitting of ISTD-PDS7.

| Class | ISTD-TR | ISTD-TR Percentage/% | ISTD-VD | ISTD-VD Percentage/% | ISTD-TE | ISTD-TE Percentage/% |
|---|---|---|---|---|---|---|
| Transverse crack | 1581 | 10.0% | 176 | 10.0% | 150 | 15.0% |
| Longitudinal crack | 921 | 5.8% | 102 | 5.8% | 126 | 12.6% |
| Alligator network crack | 236 | 1.5% | 26 | 1.5% | 249 | 24.9% |
| Patch | 763 | 4.8% | 85 | 4.8% | 88 | 8.8% |
| Pothole | 275 | 1.7% | 31 | 1.7% | 15 | 1.5% |
| Cement concrete crack | 1206 | 7.6% | 134 | 7.6% | 119 | 11.9% |
| Broken slab | 189 | 1.2% | 21 | 1.2% | 60 | 6.0% |
| Negative sample | 10,603 | 67.2% | 1178 | 67.2% | 193 | 19.3% |
| Total | 15,774 | 100.0% | 1753 | 100.0% | 1000 | 100.0% |

## 4. Baseline Methods

### 4.1. Methods

We evaluated the performance of seven state-of-the-art convolution-based and transformer-based semantic segmentation models on the ISTD-PDS7 dataset.

- **SegNet** [49]: The SegNet network achieves end-to-end learning and segmentation by sequentially using an encoder network and a decoder network. It can process input images of any size. VGG16 [67] without the fully connected layer is used as the encoding phase to achieve feature extraction. The sizes of the input images and the network parameters are reduced step by step through maximum pooling, and the pooling index position in the image is recorded at the same time. The decoding phase restores the resolution of the image through multiple upsampling. Finally, the semantic segmentation results are output by the SoftMax classifier.
- **PSPNet** [68]: The PSPNet network uses a pyramid pooling module (PPM) to aggregate contextual information from different regions to improve the ability to obtain global information. This network came first in the ImageNet Scene Parsing Challenge 2016, the PASCAL VOC Challenge 2012, and the Cityscapes test (2016).
- **DeepLabv3+** [69]: At the decoder stage, atrous convolution is introduced to increase the receptive field, and atrous spatial pyramid pooling (ASPP) is used to extract multi-scale information. DeepLabv3+ achieved a test set performance of 89% and 82.1%

without any post-processing on the PASCAL VOC Challenge 2012 and Cityscapes test (2018).

- **U-Net** [44]: The U-Net network performs skip-layer fusion for end-to-end boundary segmentation and formulates the training target with a single loss function. This model is the most commonly used model in medical image segmentation. We modified the padding mode to be the "same" so that the input and output image sizes remained the same.
- **HRNet** [70]: Differing from the above four methods, HRNet is a network model that breaks away from the traditional encoder-decoder architecture. HRNet maintains high-resolution representation by connecting high-resolution to low-resolution convolution in parallel and enhancing the high-resolution representations by repeatedly performing multi-resolution fusion across parallel convolution. In this way, it can learn high-resolution representations that are more sensitive to location.
- **Swin-Unet** [71]: Swin-Unet is a UNet-like pure transformer for image segmentation, using a transformer-based U-shaped encoder-decoder architecture with skip connections for local-global semantic feature learning.
- **SegFormer** [72]: An efficient encoder-decoder architecture for image segmentation, using multiple layers Transformer-Encoder to get multiscale features. At the same time, a lightweight multilayer perceptron (MLP) is used to aggregate semantic information of different layers. SegFormer has shown a state-of-the-art performance on the ADE20K dataset, performing better than the Segmentation Transformer (SETR) model [73], Auto-DeepLabv3+ [69], and OCRNet [74].

### 4.2. Loss Function Selection

Differing from semantic segmentation on the Pascal VOC2012 dataset [41], there are only two classes in the new dataset. Segmentation of the distress types in this paper can be seen as a DIS problem. Generally speaking, the ground-truth distress pixels stand as a minority class in the distress image (i.e., the proportion of background pixels in the new dataset is 97.42%, and the proportion of distress pixels is 2.58%), which makes it an imbalanced segmentation task. Some works [75,76] have dealt with this problem by adding larger weights to the minority class. However, in crack detection [30], it has been found that adding larger weights to the cracks results in more false positives. In order to tackle both types of imbalance during training and inference, we introduce a hybrid loss function consisting of contributions from both dice loss [77] and cross-entropy loss [78]. Specifically, the dice loss (Equation (1)) learns the class distribution, alleviating the imbalance problem, while the cross-entropy loss (Equation (2)) is used to penalize false positives/negatives while performing curve smoothing at the same time. The two loss terms are combined as shown in Equation (3), and more weight is given to the dice loss term because it can better handle the category imbalance problem. Thus, we define the pixel-wise prediction loss as shown in the following equations:

$$L(\mathbf{W})_{Dice} = 1 - \frac{2\sum_{n=1}^{N} y_n \hat{y}_n(x_n, \mathbf{W}) + \varepsilon}{\sum_{n=1}^{N} y_n + \sum_{n=1}^{N} \hat{y}_n(x_n, \mathbf{W}) + \varepsilon} \tag{1}$$

$$L(\mathbf{W})_{CE} = -\frac{1}{N}\sum_{n=1}^{N} y_n \log \hat{y}_n(x_n, \mathbf{W}) + (1 - y_n) \log(1 - \hat{y}_n(x_n, \mathbf{W})) \tag{2}$$

$$L(\mathbf{W}) = 0.9 \cdot L(\mathbf{W})_{Dice} + 0.1 \cdot L(\mathbf{W})_{CE} \tag{3}$$

Given one set of batch-size training data with $M$ input images, $S = \{(X^m, Y^m), m = 1, \cdots, M\}$, $N$ denotes the total number of pixels and is equal to the number of pixels in a single image multiplied by $x_n \in \{[1, 255], n = 1, \cdots, N\}$, where $x_n \in \{[1, 255], n = 1, \cdots, N\}$ denotes the pixel of the input image; $y_n \in \{0, 1\}$ denotes the ground-truth distress label map corresponding to $x_n$; $\hat{y}_n \in \{0, 1\}$ is the predicted probability for $x_n$ being distress or background; $\mathbf{W}$ is the set of standard parameters in the network

layers; $L(\mathbf{W})_{Dice}$ denotes the dice loss, $\varepsilon$ is used for smoothing purposes; $L(\mathbf{W})_{CE}$ denotes the cross-entropy loss; and $L(\mathbf{W})$ denotes the total loss.

### 4.3. Evaluation Metrics

In this paper, to provide a comprehensive evaluation, the *Precision*, *Recall*, *F*1, and *mIoU* are used to quantitatively evaluate the performance of the different segmentation models. For each image, the *Precision* and *Recall* can be calculated by comparing the detected distress with the human-annotated ground truth. The *F*1 ($2 \times \frac{Precision \times Recall}{(Precision+Recall)}$) is the *Precision* and *Recall* harmonic average. The intersection over union (*IoU*) reflects the overlap degree between the recognized samples of the same class and the real samples, and the *mIoU* metric is the average value of the *IoU*. The *mIoU* can be calculated as follows:

$$mIoU = \frac{1}{N}\sum_{k=1}^{N}\frac{TP_K}{TP_K + FP_K + FN_K} \tag{4}$$

where $TP_K$, $FP_K$, and $FN_K$ represent the true positives, false positives, and false negatives, respectively, $N = 2$.

## 5. Experiments and Results

In this section, we first introduce the experimental settings. The distress segmentation performance on the test dataset of the seven representative end-to-end semantic segmentation models (see Section 4.1) selected for training on the new dataset is then compared and discussed in detail. In addition, we describe the comprehensive comparative analysis conducted between ISTD-PDS7 and the open-source datasets on the optimal baseline model. Finally, the influence of negative samples and the effect of data augmentation methods on the segmentation accuracy for road pavement distress in complex scenes are analyzed. Figure 8 shows the specific experimental flowchart of this paper.

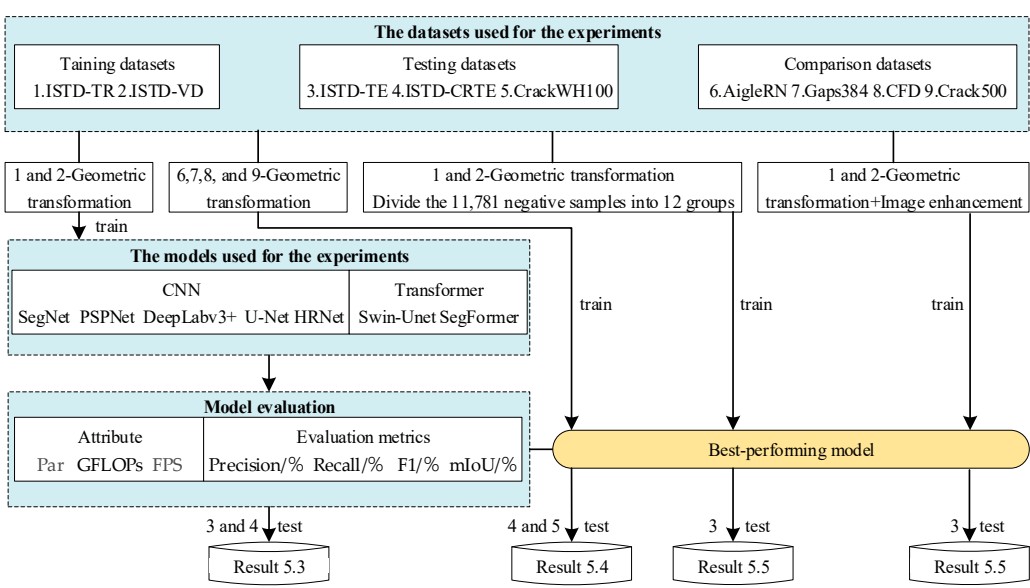

**Figure 8.** Experimental research flowchart.

### 5.1. Implementation Details

We implemented the different networks using the publicly available PyTorch, which is well-known in this community. To improve the learning performance, transfer learning [79] was adopted to train all the experimental models. The pre-trained models selected the optimal weight of the backbone network trained on the Cityspaces dataset. The output category is set to 2. The models were then trained in a freezing phase (50EP) and a thawing

phase (100EP). The batch processing size of the two phases was set to 4 and 8, respectively, and the learning rates were set to $10^{-4}$ and $10^{-5}$, respectively. The momentum and weight decay were set to 0.9 and 0, respectively. The model training process was optimized using the AdamW optimizer with a learning rate of $1 \times 10^{-4}$. All the experiments described in this paper were performed using a single GeForce RTX 3090 GPU.

*5.2. Dataset Setup*

As described in Section 3.1, the ISTD-PDS7 dataset was built from the ground up and covers a highly diverse set of geometric structures and image scenarios for pavement distress. Thus, its diversity (i.e., resolution, image features, object complexity, and markup accuracy) and distribution differ from the existing datasets.

- **Baseline evaluation of datasets:** As described in Section 3.3, ISTD-TR and ISTD-VD were used for the training and validation of the baseline evaluation networks. Data augmentation was performed to enlarge the number of distress samples in the training set, including vertical flip, horizontal flip, and flip and transpose, to balance the ratio of positive and negative samples. After the data augmentation, we obtained a training set of 30,475 images in total, containing 14,620 distress samples and 11,974 negative samples. It is worth noting that the data augmentation was not applied to the ISTD-TE or ISTD-CRTE. All the baseline models used ISTD-TE and ISTD-CRTE as the test sets to evaluate their performance in the multi-type of distress DIS task and crack segmentation task in complex scenarios.

- **Dataset comparison:** In this study, the best-performing model from the baseline assessment was used as the evaluation tool to make cross-dataset evaluations [80] of CFD [37], CRACK500 [38], AigleRN [39], and GAPs384 [38], which are labeled with the actual crack width. In order to ensure the fairness of the evaluation, we used the same data enhancement methods described above, and increased the training data size to about 14,600 pieces. CrackWH100 [30] and ISTD-CRTE were selected as the test sets. It must be noted that the images in the CrackWH100 dataset were acquired by a linear CCD camera, and we re-annotated their ground truth by the actual pixel width of the crack using LabelMe.

- **Influence of negative samples and different data augmentation methods:** We took the best-performing model in the baseline assessment as the experimental tool, randomly divided the 11,781 negative samples into 12 groups with roughly 1000 images per group and added one group of negative samples each time to participate in the model training. Then, we use two data augmentation methods, geometric transformation (vertical flip, horizontal flip, and flip and transpose) and image enhancement (shift scale rotate, random contrast, random brightness, blur, and CLAHE), on the training set to explore the influence of different data augmentation methods on the performance of the distress segmentation. Image geometric transformation simulates the change of direction and angle when an image is taken. Blur change takes into account the possible instability of the imaging camera when the light is low and the lens is unfocused. Brightness transition mainly enhances or weakens the illumination, considering the unstable situations that may occur in the case of insufficient light or severe exposure during shooting. The ISTD-TE dataset was taken as the test set in this part.

*5.3. ISTD-PDS7 Benchmark*

5.3.1. Quantitative Evaluation

Table 4 lists the overall performance of all the comparative models on the ISTD-PDS7 validation and test sets. It can be observed that the performance of the different models in the crack segmentation task is lower than that in the multi-type of distress segmentation task, because the crack structure is more complex and fine, so it requires the models to keep as much spatial information as possible, which is challenging to most models. Compared with the other models, SegFormer [72] based on the transformer

achieves the most competitive performance in the four evaluation indicators. In contrast, the mIoU values of Swin-Unet [71] on ISTD-TE and ISTD-CRTE decrease by 7.85 and 9.00, respectively. It is worth noting that although Swin-Unet has a similar number of network parameters as SegFormer, it has a low computational complexity (GFLOPs = 52.56) and high computational speed (frames per second (FPS) = 110.95). In addition, the HRNet [70] and U-Net [44] models based on convolution achieve an average performance. Compared with the other two models, HRNet [70] performs better in the distress DIS task, while U-Net [44] performs better in the crack segmentation task. The two models achieve the lowest FPS values and show a slow reasoning speed. SegNet [49], with the simple encoder-decoder architecture, performs the worst on the test sets. These baseline results will provide scientific data support for subsequent model research.

**Table 4.** Quantitative evaluation on the ISTD-PDS7 validation and test sets. V-16 = VGG16 [67], MV2 = MobileNetV2 [81], R-50 = ResNet50 [18], H-V2 = HRNetV2 [70], X = Xception [82], and ST = swin transformer [71].

| Dataset | Metric | SegNet [49] | PSPNet [68] | | DeepLabv3+ [69] | | U-Net [44] | HRNet [70] | Swin-Unet [71] | SegFormer [72] |
|---|---|---|---|---|---|---|---|---|---|---|
| Attribute | Backbone | V16 | MV2 | R-50 | MV2 | X | V-16 | H-V2 | ST | MiT-B2 |
| | Input size | 512 × 512 | 473 × 473 | 473 × 473 | 512 × 512 | 512 × 512 | 512 × 512 | 480 × 480 | 224 × 224 | 512 × 512 |
| | Par/M | 16.32 | 2.38 | 46.71 | 5.813 | 54.709 | 24.89 | 29.538 | 27.18 | 27.348 |
| | CC/GFLOPs | 601.78 | 5.28 | 118.43 | 52.87 | 166.841 | 450.602 | 79.915 | 52.56 | 113.427 |
| | Speed/FPS | 57.91 | 131.38 | 75.53 | 100.32 | 47.05 | 26.99 | 23.32 | 110.95 | 34.75 |
| ISTD-VD | Precision/% | 74.84 | 86.31 | 82.69 | 83.41 | 86.87 | 87.05 | 87.22 | 85.02 | 88.39 |
| | Recall/% | 57.12 | 82.83 | 80.2 | 80.05 | 80.43 | 87.81 | 83.66 | 77.33 | 89.69 |
| | F1/% | 64.79 | 84.53 | 81.43 | 81.70 | 83.53 | 87.43 | 85.40 | 80.99 | 89.04 |
| | mIoU/% | 54.15 | 75.72 | 72.08 | 72.64 | 74.33 | 79.47 | 76.80 | 71.3 | 81.67 |
| ISTD-TE | Precision/% | 81.14 | 92.14 | 85.67 | 89.44 | 96.31 | 92.77 | 92.80 | 89.56 | 93.64 |
| | Recall/% | 63.98 | 87.27 | 80.56 | 87.94 | 83.49 | 91.30 | 92.65 | 88.66 | 94.82 |
| | F1/% | 71.55 | 89.64 | 83.04 | 88.68 | 89.44 | 92.03 | 92.72 | 89.11 | 94.23 |
| | mIoU/% | 60.40 | 82.27 | 73.55 | 81.03 | 81.13 | 85.96 | 87.07 | 81.64 | 89.49 |
| ISTD-CRTE | Precision/% | 87.31 | 79.16 | 76.08 | 84.46 | 90.62 | 85.17 | 84.15 | 82.79 | 87.22 |
| | Recall/% | 60.45 | 68.78 | 73.11 | 79.64 | 70.42 | 86.67 | 86.02 | 76.93 | 87.06 |
| | F1/% | 71.44 | 73.61 | 74.57 | 81.98 | 79.25 | 85.91 | 85.07 | 79.75 | 87.14 |
| | mIoU/% | 57.05 | 63.50 | 65.00 | 72.66 | 67.50 | 77.50 | 76.43 | 70.12 | 79.12 |

### 5.3.2. Qualitative Evaluation

Figure 9 presents a qualitative comparison between the seven baseline methods. As shown in the first column of Figure 9, the seven kinds of pavement distress and negative sample images were randomly selected, some of which are affected by interference noise, such as shadows, oil stains, or zebra crossings. A visual inspection shows that SegFormer based on the hierarchical attention mechanism module outperforms the other six methods in the multi-type of pavement distress DIS task, especially for the distress types with different sizes, illumination, and interference noise, and the false positive predictions are reduced. For the semantic segmentation model based on convolutional operations, SegNet can only achieve a rough segmentation of the distress examples, and shows poor processing of details, such as poor continuity of the crack extraction results, and it has difficulty in extracting the planar area of pothole. In addition, Table 3 and columns 4–7 in Figure 9 show that the PPM and ASPP modules are, respectively, used in PSPNet and DeepLabv3+ for contextual information. Due to the multi-scale pooling and atrous convolution operations, a large amount of detailed information is lost, along with detailed boundaries of the pavement cracks, thereby reducing the continuity of the fine cracks. In contrast, U-Net (column 8 in Figure 9) and HRNet (column 9 in Figure 9), which take into account the fusion of multi-scale feature information, are more suitable for the task of pavement distress prediction and obtain better prediction results, but the prediction speed is the slowest (see Table 2). Columns 10 and 11 of Figure 9 demonstrate the potential of a pure transformer backbone for dense prediction tasks, compared to a CNN. In terms of operation speed, Swin-Unet obtains the second-fastest prediction speed, with 110.95 FPS, due to the low computational complexity and few parameters, but the pure transformer operation in this model results in a loss of detailed information and insufficient edge information for the crack prediction.

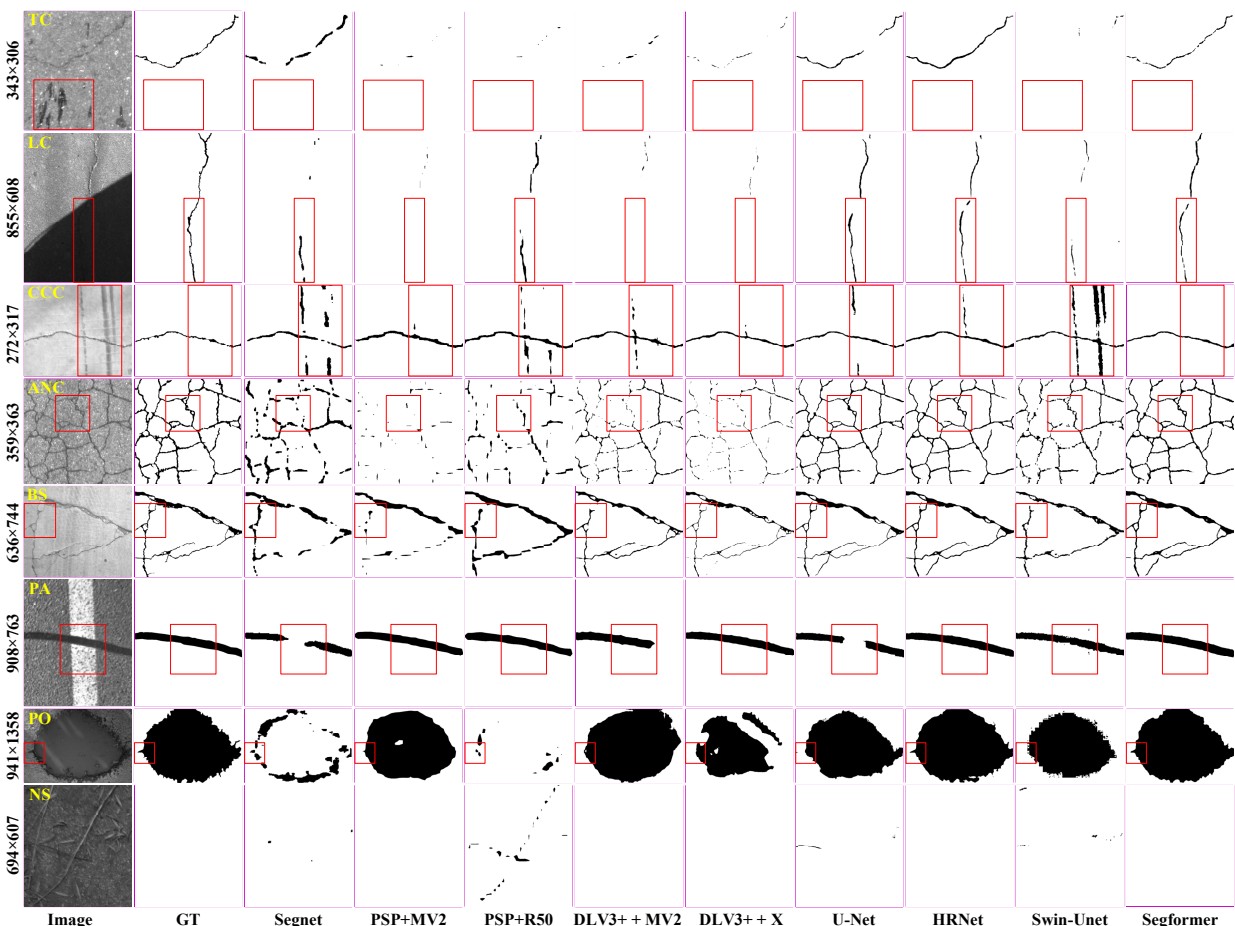

**Figure 9.** Segmentation results comparison for the seven classes of distress and distractors in complex scenes by the seven methods on the ISTD-PDS7 dataset. The red boxes in each image in the first row are the focus areas.

In contrast, a transformer encoder with hierarchical structure is used in SegFormer, which outputs multi-scale features instead of single low-resolution features, as in the vision transformer (ViT) model [83]. In addition, positional encoding is removed, which can avoid the problem of intensive prediction performance degradation caused by the interpolation of positional codes when the testing resolution differs from the training resolution [72]. The proposed MLP decoder aggregates information from different layers and combines local attention and global attention to render powerful feature representation capabilities, thus effectively reducing false positive predictions. The above findings and baseline results will provide certain reference information for the design of subsequent intelligent distress segmentation networks.

*5.4. Comparison with the Public Datasets*

Table 5 provides the quantitative comparison results of SegFormer trained by the AigleRN, CFD, CRACK500, GAPs384, and ISTD-PDS7 datasets on the two test sets. Obviously, the model trained based on ISTD-PDS7 shows the best crack extraction performance in the two test sets. Specifically, compared to the AigleRN and CRACK500 datasets. On the CrackWH100 test set, the SegFormer achieves an increase of 5.07% to 10.09 % in the F1 values and 6.17% to 11.62% in the mIoU values after being trained using the ISTD-PDS7 dataset. On the ISTD-CRTE test set, the SegFormer achieves an increase of 3.36% to 8.39% in the F1 values, and 25.65% to 5.35% in the mIoU values after being trained using the ISTD-PDS7 dataset. In addition, the results of Table 5 show that the SegFormer trained using the four public datasets, F1, and mIoU scores on the CrackWH100 test set were better

than those on the ISTD-CRTE test set. According to a comparison of the two test sets, it was found that the crack images in the ISTD-CRTE contain more texture similarity noises, such as shadows, branches, stains, etc. (Figure 5). The new test set also includes dramatically changed illumination conditions and various crack width sizes (Figure 4), which were more complex than the existing public datasets. Consequently, when evaluating the crack detection performance of different crack detection models in complex scenarios, it can be much more challenging to use it as the test set.

**Table 5.** Performance comparison for SegFormer trained on the different datasets.

| Dataset | Metric | AigleRN [39] | Gaps384 [38] | CFD [37] | Crack500 [38] | ISTD-PDS7 |
|---|---|---|---|---|---|---|
| CrackWH100 [6] | Precision/% | 78.38 | 76.75 | 75.48 | 84.29 | 85.73 |
| | Recall/% | 80.36 | 87.28 | 93.83 | 84.47 | 93.50 |
| | F1/% | 79.36 | 81.68 | 83.66 | 84.38 | 89.45 |
| | mIoU/% | 70.58 | 72.32 | 73.39 | 76.03 | 82.20 |
| ISTD-CRTE | Precision/% | 89.86 | 71.65 | 75.72 | 78.16 | 87.22 |
| | Recall/% | 55.67 | 84.58 | 88.81 | 90.26 | 87.06 |
| | F1/% | 68.75 | 77.58 | 81.74 | 83.78 | 87.14 |
| | mIoU/% | 53.47 | 66.40 | 71.12 | 73.77 | 79.12 |

Figure 10 shows the crack detection results obtained on the two test sets. A visual inspection shows that SegFormer trained with the five different datasets shows a different performance on the two test sets, and the model trained on ISTD-PDS7 performs better in the crack detection tasks than the others, especially for cracks with different appearances, contrast, and interference conditions. Specifically, as shown in Figure 10a,f, although the cracks in the original images are affected by water stains and shadows, the model trained with the new dataset can still effectively depict cracks under the low contrast condition, while the models trained with the other datasets can extract cracks in the rectangular box area with a low degree of confidence. In addition, as shown in Figure 10b,c, the four public dataset trained models misclassify grass roots/branches (which are highly similar to the crack texture) as cracks to generate false positive predictions. It is worth noting that, as shown in Figure 10d, when an image with a "white crack" exists, the model trained with the new dataset can correctly and completely delineate the crack, while the models trained based on the other datasets fail since they are lacking in "white crack" samples. It can also be observed that, as shown in Figure 10e, for the alligator network crack with a clear appearance, the continuity of the cracks delineated by the model trained on the AigleRN dataset is the worst, which is also consistent with the result for the lowest image complexity of this dataset (see Table 2). The models trained with the other datasets produce more complete predictions, but the model trained with the new dataset shows more detailed crack extraction results. Thus, it can be concluded that training the model on the ISTD-PDS7 dataset can effectively improve the precision of crack extraction and the robustness of background noise suppression.

*5.5. Influence of Negative Samples and Data Augmentation Methods*

**Influence of Negative Samples:** The ISTD-PDS7 dataset includes more negative samples that are highly similar to pavement distress in terms of gray-level and texture characteristics. In this experiment, SegFormer was used as the experimental model (see Section 5.3 for details) to analyze the influence of negative samples on the segmentation accuracy of pavement distress in complex scenes. Figure 11 shows the variation trend of F1 and mIoU on the ISTD-TE dataset with the increase in the number of negative samples involved in the training.

As can be seen from Figure 11, with the increase in the negative samples, the segmentation performance of the trained model is gradually improved. When all the negative samples participate in the model training, the F1 and mIoU are increased by 4.25% and 7.00%, respectively. When the number of negative samples participating in the training

reaches 5000, the difference between the negative sample images participating in the training and the overall negative samples in the ISTD-PDS7 dataset is significantly reduced, resulting in the growth trend of the F1-score and mIoU of the subsequent model slowing down, but the model does not reach a saturation state.

Figure 12 visually demonstrates the effectiveness of negative samples in reducing the false extraction of road interference noise in complex scenes. When the original CCD imagery contains complex interference noise, such as shadows, oil stains, water stains, branches, scratches, and tire indentation (Figure 12, line 1), if negative samples are not used in the model training, SegFormer can overcome some of the noise interference, but in the face of interference noise that is very similar to the pavement distress characteristics, there are still many false positive predictions (Figure 12, line 2). In contrast, when the number of negative samples involved in the model training reaches 5000, the false positive predictions decrease significantly (Figure 12, line 3). When all the negative samples participate in the model training, most of the false positive predictions disappear (Figure 12, line 4). Therefore, it can be concluded that only using normal samples in the training cannot achieve a satisfactory precision. However, after the introduction of the targeted negative samples in the model training, the DCNN can extract richer semantic information from the imagery. By judging the gap between features, the network focuses on the distress areas, which makes the model have a higher feature expression ability and effectively reduces the false extraction of complex interference noise. In order to meet the practical engineering requirements, in our future dataset maintenance and algorithm research, we will expand the negative samples through the developed manual inspection software.

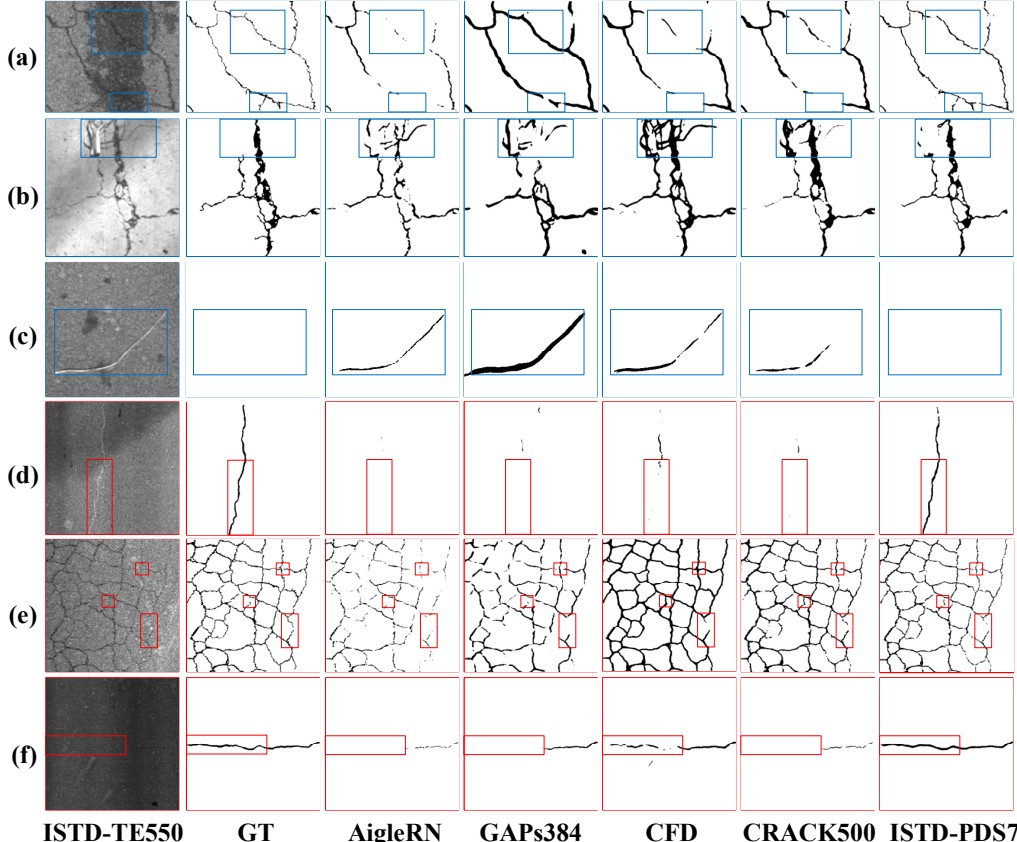

**Figure 10.** Comparison of the segmentation results of SegFormer after training on different datasets, (**a**–**c**) come from ISTD-PDS7; (**d**–**f**) are come from CrackWH100. The boxes in each image are the focus areas.

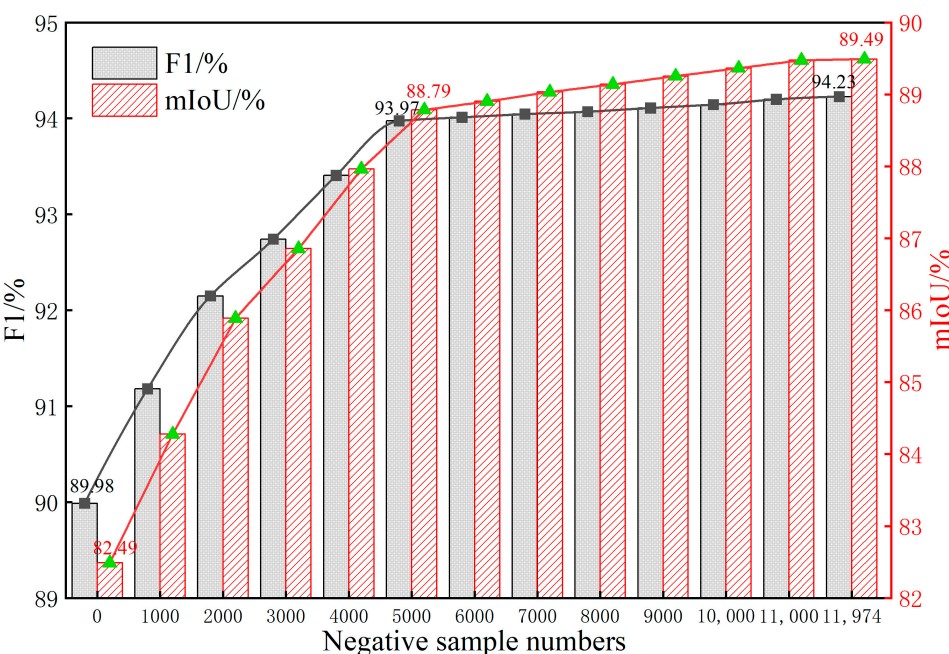

**Figure 11.** Effects of different numbers of negative samples on the distress segmentation performance.

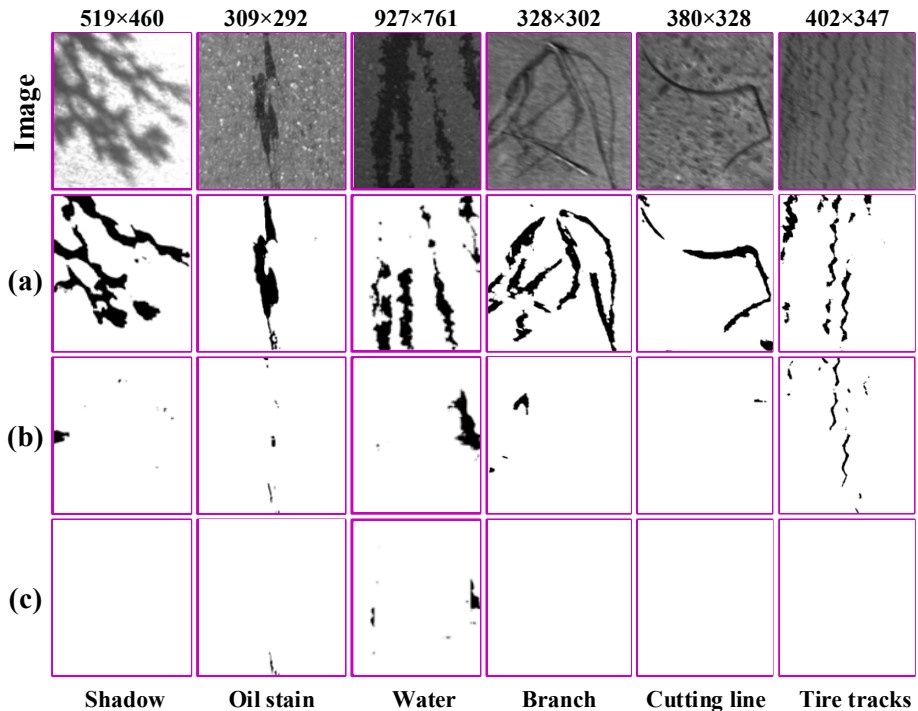

**Figure 12.** The influence of negative samples on fracture extraction. (**a**) SegFormer prediction results when negative samples are not used in the training. (**b**) SegFormer prediction results when 5000 negative samples are used in the training. (**c**) SegFormer prediction results when 11,781 negative samples are used in the training.

**Influence of Data Augmentation Methods:** In computer vision, image augmentation has become a common implicit regularization technique to solve the overfitting problem in DL models and is widely used to improve performance. In this experiment, SegFormer was used as a test model to verify the impact of two data augmentation techniques.

According to the quantitative comparison in Table 6 overall, the two data augmentation techniques adopted on the ISTD-PDS7 dataset can effectively improve the performance

of SegFormer. While it also can be seen from Table 6 that the most direct geometric transformation improves the mIoU by 5.60%, the image enhancement results in a relatively small improvement in the distress segmentation performance, and the mIoU is improved by only 1.90%. ISTD-PDS7 retains the two types of enhancement methods and can improve the mIoU by 6.79%. The geometric transformation technique increases the variation of the distress, such as the position, viewpoint, scale, etc. These operations can simulate multiple variations of objects in natural scenes, thus improving the within-class richness. Although the image enhancement method also increases the richness of the imagery, the reason for its small performance improvement may be that, although the boundary features of the distress in the original image are changed, the corresponding label is not changed.

**Table 6.** Results of different data augmentation techniques on the test set.

| Method | Training Set | Precision/% | Recall/% | F1/% | mIoU/% |
|---|---|---|---|---|---|
| None | 17,147 | 92.04 | 89.3 | 90.65 | 83.89 |
| Image enhancement | 47,601 | 92.21 | 91.63 | 91.92 | 85.79 |
| Geometric transformation | 30,475 | 93.64 | 94.82 | 94.23 | 89.49 |
| All | 60,929 | 93.92 | 95.74 | 94.82 | 90.68 |

Furthermore, Figure 13 shows the influence of the two types of data augmentation techniques adopted in this study on the details of the distress segmentation. In comparison, it is found that both types of data enhancement technique improve the accuracy and completeness of the distress segmentation.

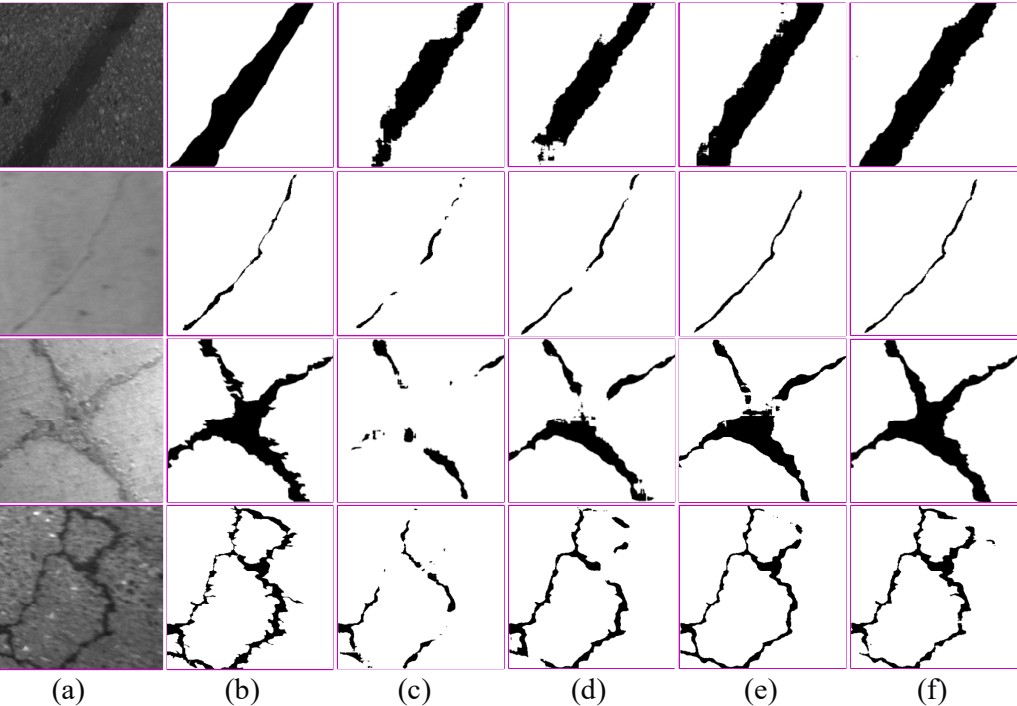

|  | (a) | (b) | (c) | (d) | (e) | (f) |

**Figure 13.** The segmentation performance with different data augmentation techniques. (**a**) The original image of road pavement distress. (**b**) The ground truth. (**c**) The segmentation results without data augmentation. (**d**) The segmentation results using only image enhancement. (**e**) The segmentation results using only geometric enhancement. (**f**) The segmentation results of using the two data augmentation techniques.

## 6. Conclusions

In this paper, we first reviewed the related research on the automatic detection of highway pavement distress based on DL techniques. We found that the current status of

the public datasets and test results is insufficient for engineering-level application research, which seriously limits the progress of automatic pavement distress segmentation in complex scenes. Therefore, to solve the problem, we built a new large-scale DIS dataset for multi-type pavement distress segmentation, i.e., the ISTD-PDS7 dataset, based on measured highway pavement CCD images, which is currently the largest and most challenging dataset for the semantic segmentation of pavement distress. Additionally, we evaluated a set of representative semantic segmentation methods on the new dataset, which can serve as baseline results for future works. Quantitative assessments and qualitative inspections demonstrate that the Segformer model with multiple layers Transformer-Encoder is more suitable for the segmentation of multiple types of pavement diseases in complex scenes. The ISTD-PDS7 dataset can effectively improve the precision of crack extraction and the robustness of background noise suppression when compared to other publicly accessible datasets. The inclusion of negative samples in model training can effectively avoid false positive detection of models. Additionally, the necessary data augmentation methods can considerably improve the pavement distress segmentation performance of the model. Furthermore, based on the new dataset, we intend to carry out additional research on data augmentation methods, training strategies, and model improvement tactics to promote the development of applications in this field.

**Author Contributions:** Conceptualization, W.S.; data curation, Z.Z., H.Z. and J.Z.; formal analysis, Z.Z., B.Z. and G.J.; funding acquisition, W.S.; investigation, G.J.; methodology, W.S. and Z.Z.; project administration, W.S.; software, H.Z.; visualization, J.Z.; writing—original draft, W.S. and Z.Z.; writing—review and editing, B.Z., G.J., H.Z. and J.Z. All authors have read and agreed to the published version of the manuscript.

**Funding:** This work was funded in part by The National Natural Science Foundation of China under Grant 42071343 and 42204031, and in part by the Basic Scientific Research Expenses of Heilongjiang Provincial Universities, China, under Grant 2020-KYYWF-0690.

**Data Availability Statement:** The ISTD-PDS7 dataset can be obtained on request by e-mail from https://ciigis.lntu.edu.cn/, accessed on 1 October 2023.

**Conflicts of Interest:** The authors declare no conflict of interest.

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
