# Peer review of "ISTD-PDS7: A Benchmark Dataset for Multi-Type Pavement Distress Segmentation from CCD Images in Complex Scenarios"

_remotesensing, doi:10.3390/rs15071750_

Round 1

Reviewer 1 Report

The ISTD-PDS7 dataset contains 18,257 images of pavement cracks, covering seven types of pavement damage in nine scenes. This dataset is mainly used for pavement disease identification based on semantic segmentation, including asphalt pavement and concrete pavement. The dataset is comprehensive and specific in classification. The author carried out quantitative and qualitative experiments on the pavement disease dataset at this stage, which fully proved the comprehensiveness and superiority of ISTD-PDS7 dataset.

However, there are the following questions about the content of the paper:

(1)Whether the ISTD-PDS7 dataset contains pavement images without diseases, so as to avoid false positive detection for the model of detecting road damage.

(2)The resolution of the collected image is 3517px×2193px, and the image size of the actual data set is about 375px×371px. The information contained in the cropped image will be affected. Has the author considered this issue? During the testing, NHA12D adopted the method of testing by segments. It is suggested that the author pay more attention.

(3)It is mentioned in the abstract that two data enhancement methods have been developed to improve the segmentation accuracy. In Section 5.2, ISTD-PDS7 dataset is expanded by geometric changes and image enhancement. However, the article does not introduce the improvement of the two data enhancements in detail. At this stage, there are also methods to enhance the data by GAN, and it is suggested that the author pay more attention to it.

Author Response

请参阅附件。

Reviewer 2 Report

The paper builds the large-scale dichotomous image segmentation dataset for multi-type pavement distress segmentationThe dataset covers seven types of pavement distress in nine types of scenariosIt makes a certain contribution to the pavement distress segmentation. However, there are still some shortcomings in this paper. My detailed comments are as follows:

1There is a spelling error in the text. Please check the full text. For example, "was builted."

2The description of the data volume of the dataset in the paper is inconsistent. The amount of dataset is 18527 frames in the abstract, but the amount of labeled data is 17527 frames in the Introduction, and the amount of data is different between the two places. Please give a reasonable explanation. The statement is as follows "The final dataset contains 18,527 images, which is many more than the previously released benchmarks. " And "The main contributions of this paper can be summarized as the following three aspects: 1. A large-scale extendable DIS dataset—ISTD-PDS7—containing 17,527 CCD images and seven types of pavement distress, was builted and annotated manually by four experts in the field of pavement distress detection. "

3The data set established in this paper only contains one category per frame. What happens when an image contains multiple pavement distress?

4The number of negative samples in the dataset is twice the number of labeled positive samples. Whether such a data amount configuration affects the data balance and is not conducive to model training.

5The article mentions some novel evaluation metrics for dataset evaluation. What are these evaluation metrics?

6The article mentioned that the data set established in this paper has higher accuracy of annotation, how to judge the accuracy of annotation, qualitative judgment?

Author Response

请参阅附件。

Reviewer 3 Report

The article is a topical research and is devoted to building a dichotomous image segmentation (DIS) dataset for multi-type pavement distress segmentation. The scientific novelty and the significance of the work is high. There are some prospects for further developments. While the authors' dataset performs well, there is still a long way to go before it becomes more usable for engineering applications. The article is strong from an empirical and applied point of view. Large scale experimental studies have been carried out. The authors have demonstrated excellent skills in the applied methods. Their scientific approach is interesting. 

COMMENTS

1) In my opinion, the article is too long and could be divided into two shorter articles. First, about creating the benchmark dataset ISTD-PDS7 and comparisons with other datasets. Second article could be about benchmarking selected deep learning models using the dataset from the first article.

2) What is the origin of the name of this benchmark dataset?

3) It is advisable at the beginning of the Experiments section to draw up a program of experimental studies in the form of a flowchart.

Author Response

请参阅附件。

Reviewer 4 Report

This paper proposes a dataset for pavement distress segmentation. The dataset covers seven types of pavement distress in nine types of scenarios, with negative samples. The authors should be commended for developing a dataset and baseline to encourage more research on the topic. However, the literature search and references could be expanded to include recent developments in civil engineering or transportation/pavement engineering areas. The conclusion part should be revised to reflect research findings.

Some questions and comments for the authors to consider for possible improvements:

1.      Line 332. Authors need to elaborate on why the proposed dataset has ‘higher labeling accuracy’.

2.      Line 397. Authors need to specify how to further split the ISTD-TE to obtain ISTD-CRTE. What is the difference between ISTD-TE and ISTD-CRTE?

3.      Line 491. The authors need to specify the transfer learning settings, which could be one of the major contributors to the model’s performance. Also, the authors should be aware that using transfer learning to prevent models from overfitting is only effective with small datasets.

4.      Line 561. The statement has nothing to do with Figure 8.

5.      Line 594-599. The description of the comparison is confusing.

6.      Line 602. The authors need to elaborate on why the ISTD-CRTE is more useful for evaluating model robustness.

7.      The idea of using ISTD-CRTE as test dataset to evaluate public datasets is debatable. While ISTD-CRTE is homogeneous with ISTD-PDS7, it is expected that the model trained on ISTD-PDS7 could yield better performance.

8.      The author should proofread the whole paper thoroughly. Such as follows:

Line 90: wrong statement

Line 330: wrong figure caption

English editing and proofreading is recommended

Author Response

请参阅附件。

Round 2

Reviewer 2 Report

All my queries were addressed successfully.However, it is recommended to check the correct use of tenses in articles.